# Lung Cancer (LC) in HIV Positive Patients: Pathogenic Features and Implications for Treatment

**DOI:** 10.3390/ijms21051601

**Published:** 2020-02-26

**Authors:** Stefano Frega, Alessandra Ferro, Laura Bonanno, Valentina Guarneri, PierFranco Conte, Giulia Pasello

**Affiliations:** 1Medical Oncology 2, Istituto Oncologico Veneto IOV- IRCCS, 35, 128 Padova, Italy; stefano.frega@iov.veneto.it (S.F.); alessandra.ferro@iov.veneto.it (A.F.); laura.bonanno@iov.veneto.it (L.B.); valentina.guarneri@unipd.it (V.G.); pierfranco.conte@unipd.it (P.C.); 2Department of Surgery, Oncology and Gastroenterology, University of Padova, 35, 128 Padova, Italy

**Keywords:** HIV, PLWH, lung cancer, chemotherapy, immunotherapy

## Abstract

The human immunodeficiency virus (HIV) infection continues to be a social and public health problem. Thanks to more and more effective antiretroviral therapy (ART), nowadays HIV-positive patients live longer, thus increasing their probability to acquire other diseases, malignancies primarily. Senescence along with immune-system impairment, HIV-related habits and other oncogenic virus co-infections increase the cancer risk of people living with HIV (PLWH); in the next future non-AIDS-defining cancers will prevail, lung cancer (LC) in particular. Tumor in PLWH might own peculiar predictive and/or prognostic features, and antineoplastic agents’ activity might be subverted by drug-drug interactions (DDIs) due to concurrent ART. Moreover, PLWH immune properties and comorbidities might influence both the response and tolerability of oncologic treatments. The therapeutic algorithm of LC, rapidly and continuously changed in the last years, should be fitted in the context of a special patient population like PLWH. This is quite challenging, also because HIV-positive patients have been often excluded from participation to clinical trials, so that levels of evidence about systemic treatments are lower than evidence in HIV-uninfected individuals. With this review, we depicted the epidemiology, pathogenesis, clinical-pathological characteristics and implications for LC care in PLWH, offering a valid focus about this topic to clinicians.

## 1. HIV and Cancer Risk

The human immunodeficiency virus (HIV) infection remains an important global health issue. Despite a progressive decrease of new cases of HIV infection in Europe, this reduction is far from the target identified by the WHO for 2020. Data from Global HIV/AIDS statistics evidence that in 2018, worldwide estimation of people living with HIV (PLWH) was 37.9 million [32.7–44.0 million] with 1.7 million people becoming newly infected in 2018 globally. Among them, about 23.3 million were accessing antiretroviral therapy (ART), up from around 8 million in 2010. Thanks to highly active antiretroviral therapy (HAART), the morbidity and mortality of HIV-infected individuals reduced in last years, indeed, since the peak in 2004, AIDS-related deaths are almost halved [1].

PLWH are at higher risk for cancer than common people, suggesting that HIV is an independent risk factor for its development [2]. In the initial phase of the AIDS epidemic, non-Hodgkin lymphoma (NHL), cervical premalignant lesions and Kaposi sarcoma (KS) were the prevalent form of cancer, and connected to immune-deficient status of PLWH [3]. Accordingly, in 1990, the US Centers for Disease Control defined these malignancies as AIDS-defining cancers (ADCs) [4].

With the development in medical care for HIV infection and the widespread use of HAART, the outcome of PLWH has dramatically improved, in part also as a consequence of a decreased rate and mortality from different HIV-related illness, such as ADCs [5]. As this regard, data from the USA registers, showed that Kaposi sarcoma and NHL cases dropped by more than 80% and 50% from 1991–1995 to 2001–2005, respectively [6]. However, PLWH still remain at higher risk for ADCs in comparison to the general population, according to recent studies [7,8].

Improving life expectancy, the spectrum of malignancies occurring in PLWH changed substantially in last years: in particular, the incidence of several non-AIDS-defining cancers (NADCs) such as lung cancer (LC), Hodgkin’s disease, anal carcinoma, skin cancer, gastrointestinal carcinoma were found to be higher in PLWH than in people without infection [9]. Especially in developed countries, the leading cause of deaths in PLWH has gradually become cancer, and predominantly NADCs in place of ADCs [10]. Moreover, forecasts for the next years provide for LC and prostate expected to emerge as the most common malignancies types by 2030 [11].

## 2. Lung Cancer Features in HIV Infected Patients

LC is both the leading NADCs and the main cause of death from cancers among HIV-positive people [12]. Incidence of LC in HIV-positive patients is remarkable; standardized incidence ratios (SIRs) for LC resulted more than 2.5 higher in PLWH in comparison to the general population, according to two meta-analyses [3,13].

The mechanisms driving increased LC risk among PLWH can be several: (I) tobacco smoking is a quite frequent habit, usually higher than in HIV-negative individuals [14,15]; (II) HIV infection leads to a chronic inflammation and often increases the risk of other viruses co-infections, thus negatively modulating the immune-system [16,17,18]; and (III) today life-expectancy of PLWH has improved, but elderly increases the risk to cancer due to immune-senescence [19] (Figure 1). It is important to specify that, independent of smoking, PLWH are estimated to have at least two-fold higher risk than HIV-negative individuals to develop LC [20].

Primary prevention would be extremely important, since smoking reduces life expectancy of PLWH because of the higher rate of death due to cardiovascular diseases and NADCs [21]. In particular, LC in PLWH represents nearly a third of deaths from malignancies and about 10% of non-HIV-related deaths causes [22,23].

Anticancer surveillance of PLWH fails in both humoral responses and in T-cell mediated cytotoxicity (Figure 1). Beyond tobacco smoke and immune system impairment, other risk factors may be implied in NADCs genesis. Age represents one of main single risk factor for cancer in the general population. Elderly and, in particular, the time passed since the HIV infection, raise the cancer risk of PLWH [24,25]. A possible teratogenic potential of HAART was supposed but, despite a few reported associations between certain classes of HAART drug and cancer, future studies are necessary to establish a direct carcinogenic role [26,27].

Regarding clinical characteristics, the disease is often diagnosed in locally advanced/metastatic stage in PLWH and lung adenocarcinoma (LADC) is the most common histological subtype [28].

HIV-infection status seems not to correlate with histotype and disease stage of LC [29,30] and, while controversial data in this regard have been reported [31], even more recent large cohort studies confirmed that distributions of the cancer stage at presentation and histologic subtypes are similar by HIV status [32,33].

PLWH seem to be younger when diagnosed with LC, in comparison with HIV-negative people, probably because they tend to develop malignancies earlier [32]. The HIV-linked pathogenic patterns of cancer initiation and progression, as well as the greater medical surveillance of HIV-positive people *versus* those HIV-negative might be the factors at the basis of this earlier age at LC diagnosis.

Concerning the molecular features, Thaler et al. found that LADC tumor cells of HIV-positive patients (*n* = 55) express *EGFR* or *KRAS* mutations not differently in terms of frequency when compared with those from HIV-negative patients (*n* = 136) [34]. The existence of HIV infection should not negatively influence the search for other druggable targets, such as anaplastic lymphoma kinase (ALK) rearrangement, already described in PLWH affected by non-small cell lung cancer (NSCLC) [35].

It is quite unexplored whether peculiar carcinogenic pathways promote LC initiation and progression in HIV-positive patients. The up-to-date most important contribution comes from Zheng et al., who performed a sophisticated genome-wide analysis on LC specimen by HIV-positive patients (*n* = 59). Proming 1 (PROM1), sineoculis homeobox homolog 1 (SIX1) and transcription factor AP-2 alpha (TFAP2A) genes have been found as overexpressed in LC cells from HIV-positive patients, while synaptopodin-2 (SYNPO2), alcohol dehydrogenase 1B (ADH1B) and indolethylamine N-methyltransferase (INMT) genes resulted as repressed in HIV-associated LC [36]. All these genes are known to be altered in LC tumor-initiating cells, as well as in tumorigenicity through epithelial–mesenchymal transition (EMT) pathway and metastasization process, and these findings may be useful to improve the diagnostic–therapeutic pathway of HIV-associated LC [37,38].

Thus, both etiopathogenesis and prognosis of LC in PLWH might be influenced by a differential expression of key oncogene and tumor suppressor signaling networks. Furthermore, specific immune-suppressor proteins encoded by HIV lentivirus, such as the negative regulator factor (NEF), prevent lung cell apoptosis, thus facilitating cancer initiation [39].

Different cohort studies and a meta-analysis showed that HIV-infection is a negative prognostic factor for different types of malignancies, including LC [32,40,41,42,43]. Possible explanations of this historical worst prognosis may be traced in the HIV-linked immunosuppression, as well as major healthcare disparities [28,44], both however improved in last years, thanks to a more active HAART regimen and a wider access to oncologic care. The major above-mentioned HIV-linked LC peculiarities are briefly reported in Table 1, while implications for LC treatments by HIV status are discussed in detail below.

## 3. Therapy

### 3.1. Surgery and Radiotherapy

Surgical resection is a milestone component in the therapeutic algorithm of early and locally advanced stage LC, but its safety is unclear in HIV-infected patients. A comparative cohort study evaluating LC surgical outcomes in HIV-positive patients (*n* = 22) compared to patients with negative or unknown HIV status (*n* = 2430) demonstrated more frequent surgical complications and poorer post-surgical survival [45].

However, there are conflicting data about this topic. Horberg et al. conducted an extensive observational cohort analysis of surgically treated HIV-infected patients (*n* = 352), comparing surgical outcomes and morbidity with those in matched HIV-uninfected patients. Authors did not find a statistically different complication rate between the two groups, except for incidence of postoperative pneumonia and potential 12-month mortality [46].

Moreover, a recent data collection about a national cohort of LC patients showed that surgery is safe in HIV-positive patients (*n* = 137), being surgical complications and 30-day mortality similar to those uninfected ones (*n* = 8234) [47]. Small single institution series of NSCLC patients (*n* = 6) who underwent lobectomy or segmentectomy seems to confirm the feasibility of surgery in HIV-positive patients [48].

There are no prospective data on the efficacy of thoracic radiotherapy in HIV+ patients with locally advanced LC, with available data coming from case-control series and case reports. The potentially higher toxicity risk in PLWH would be turn down with the modern technique of intensity modulated radiotherapy (IMRT) [49]. Thus, in the absence of conclusive data, there is no reason to exclude HIV-positive patients with LC from radiotherapy, when indicated. However, clinicians should pay particular attention to the management of radiation treatment side effects, especially in terms of severe esophagitis: this risk might be higher in HIV-positive patients, with CD4+ T cell count and combined HAART regimens as possible causes [49].

A recent case-control study based on Surveillance, Epidemiology, and End Results (SEER) cancer registry dataset investigated whether prognosis of elderly patients (≥65 years) affected by early-stage solid cancer is influenced by HIV-infection. PLWH affected by colorectal, breast (BC) or prostate cancer (PC) were found to have a worse outcome, even in terms of cancer-specific mortality for BC and PC, but prognosis for LC resulted not statistically different by HIV status [50].

This supports the trend over the years for a better prognosis of HIV-patients with LC, when treated similarly to those HIV-negative patients.

### 3.2. Chemotherapy and Target Therapy

Anticancer therapy deserves some caution in patients with HIV infection, because of drug-drug interactions (DDIs) and the immunosuppressive nature of the cytotoxic agents [51,52].

Several drugs currently employed in HAART regimens, in particular protease inhibitors (PIs), nucleoside reverse transcriptase inhibitors (NRTIs) and non-nucleoside reverse transcriptase inhibitors (NNRTIs), can cause drug interactions inducing or inhibiting the transporters and cytochrome (CYP) P450 enzymes, such as P-glycoprotein [53]. Indeed, many antineoplastic agents are metabolized by the CYP system, and as a consequence the combination of ART and chemotherapy could result in either a drug accumulation and excess of toxicity or vice versa in a quick drug clearance and impaired activity [51,54].

Among the PIs, ritonavir is a stronger inhibitor of CYP3A activity, while NNRTIs can induce metabolism, reducing the efficacy of antiblastic drugs.

Below we propose some example of DDIs between ART and antiblastic agents used in solid tumor, specifically in LC, in order to avoid cumulative adverse events.

The exposure to and action of antimetabolites such as gemcitabine, antitumor antibiotics and platinum is scarcely influenced by a HAART, because these agents are mainly metabolized though a route different from CYP450 [52]. Meanwhile, DDIs can be expected with alkylating agents, taxanes, vinca alkaloids, epipodophyllotoxins and corticosteroids, often used in cancer patients [55].

Specifically, paclitaxel and docetaxel are CYP2C8 and CYP3A4 substrates and their combined administration with CYP inhibitors causes an altered pharmacokinetics with an associated risk of severe myelosuppression and peripheral neuropathy. Significantly, ritonavir and ketoconazole resulted in an even riskier increase in docetaxel AUC, when administered together [56].

Moreover, vinca alkaloids, such as vincristine, are metabolized by CYP3A4, thus a combination of drugs like ritonavir or other PIs can induce high vinca levels with an increased risk of neurotoxicity, myelosuppression; ritonavir/lopinavir could increase the vincristine adverse event of paralytic ileus [57]. Epipodophyllotoxins are metabolized by the CYP3A4 so that the interaction, for example, of etoposide with a PIs could increase its blood concentration with a high-risk and severity of mucositis, myelosuppression and alteration of transaminases [54].

Platinum-based regimens have no specific interaction with ART, but cisplatin-induced nephrotoxicity may need dosage adjustment for some antiretroviral drugs, in particular for tenofovir [58].

In addition, corticosteroids are often part of premedication in chemotherapy regimens and the interaction with antiretroviral-agents could mediate a modulation of their biotransformation. In particular, dexamethasone and methylprednisolone are both substrates and concentration-dependent inducer of CYP3A4 so their blood levels may decrease the efficacy of NNRTIs and PIs [59].

Even if evidence is based on few experiences, it indicates that PLWH with malignancies should continue HAART during chemotherapy, in order to face the hematologic toxicity, that is the decrease in the number of CD4 lymphocytes and the related complications such as increasing the risk of opportunistic infections. In this regard, the prophylaxis for Pneumocystis jiroveci and other opportunistic infections should be administrated in PLWH with cancer when initiating treatment, regardless of CD4+ T-cell count [60].

Furthermore, patients who receive a combination of antineoplastic agents and HAART can achieve better response and survival rates than patients who receive chemotherapy alone [6,55].

Thanks to the progress of both HAART and anticancer therapy, the outcome for PLWH with cancers greatly improved: compared with pre-HAART era, the 5-year overall survival (OS) rate for HIV-infected patients with HL and DLBCL over time has more than doubled [10]; prognosis of patients affected by anal cancer is similar between PLWH and the general population [61]. Despite the modern HAART era, in a study conducted on 71 PLWH and 2463 HIV-uninfected individuals diagnosed with NSCLC, among PLWH OS was significantly worse in PLWH with 5-year OS probability of 9.1% and 17.9% for PLWH and HIV-uninfected individuals, respectively (*p* < 0.01) [62].

Even if the incidence of LC harboring EGFR-sensitive mutations is similar in PLWH and in the general population, data about tolerance, response to targeted therapy with tyrosine kinase inhibitors (TKIs) and prognosis are lacking. A case report of 2 HIV-positive patients diagnosed with EGFR mutated LADC showed the promising effectiveness and safety of EGFR-TKIs concomitant with antiretroviral therapy for an extended period [63].

Due to the TKIs’ extensive metabolism by CYP3A4, concomitant ART medications can increase their exposure, modifying the metabolism and efficacy of the drug [64]. Moreover, both PIs and TKIs can cause QT prolongation, arrhythmias and sudden death [58].

Experience from two HIV-infected patients with ALK-rearranged NSCLC proved that these patients had good responses to first as well as to second generation ALK-inhibitors, but both patients required a preventive modification of HAART regimens in order to avoid DDIs: ALK-TKIs are metabolized by CYP3A4, so a co-administration of CYP inhibitors can cause QT prolongation [35].

### 3.3. Immunotherapy

Immune response modulation through the use of specific immune check-point inhibitors (ICIs) has revolutionized the treatment of different cancer types in the last years.

One of the most studied mechanism driving tumor immune escape resides in the signaling pathway, involving the programmed death 1 (PD-1) expressed on the surface of activated T cells and its ligand (PD-L1) expressed by tumor cells. Immune checkpoints have become of great importance as a therapeutic target in NSCLC, with several anti PD-1 and anti PD-L1 agents being successfully tested and approved by regulatory drug agencies [65,66,67,68,69].

HIV virus infection in cancer patients may shape and interact with tumor-immune microenvironment (TiME), thus potentially affecting the successful use ICIs [70]. Chronic viral infection and cancer are pathological conditions dominated by chronic inflammation and an insufficient antigen clearance. The consequence is a difficult construction in HIV patients of a functional anti-cancer immune response, with T cells becoming exhausted for a high acquired expression of different negative check-point receptors [71] (Figure 2).

At this regard, PD-L1 expression and TiME have been investigated even in PLWH affected by LC. HIV status seems not to influence the expression of PD-L1 in NSCLC [72], even if in some case-series HIV-infected patients have found to exhibit a higher PD-L1 expression [73]. Significantly, immune-cell infiltration was found to be higher in HIV tumors, characterized by more CD8+ T cells, B cells and macrophages than in control patients. These findings suggest that PLWH might be equally responsive to ICIs [73].

Moreover, in a small case-control study of 26 LC patients, tumor cells from HIV-positive subjects (*n* = 13) express much more the receptor B7-H3 expression compared to HIV-negative ones (*n* = 13), while tumor PD-L1 expression, as well as PD-1 and PD-L1 expression on tumor infiltrating lymphocytes (TILs) was similar between the two groups [74]. B7-H3 could negatively regulate T-cells activation and tumor-associated macrophages polarization, thus promoting cancer immune escape [75].

No difference in tumor PD-L1 expression status by HIV status was found, also according to another cohort study, that showed how a high PD-L1 level in NSCLC resulted to be a negative prognostic in HIV-positive but not in HIV-negative individuals. Thus possibly, in those HIV patients, the PD-1/PD-L1 interaction exerts an immune suppressive action more pronounced than in HIV-uninfected ones [76].

The efficacy of PD-1/PD-L1 blockade therapy will likely depend on the potency of cytotoxic effect of CTLs. CTLs response in PLWH is characterized by severe exhausted CD8+ due to persistent immune stimulation by HIV [77], so that antitumor activity of CTLs might be weak even after blockade of the PD-1/PD-L1 pathway.

Anyhow, recent analyses showed preliminarily a reactive TiME that characterize NSCLC from HIV-positive patients immune-reactive microenvironment in HIV-associated NSCLC [73,78].

A recent review involving patients (*n* = 73) treated with ICIs targeting PD-1/PD-L1 axis and/or CTLA-4 showed that ICIs seem safe and active in HIV-positive patients suffering from advanced solid cancer [79].

Evidence on this issue comes also from results of the first prospective study, a phase I trial evaluating the safety profile of pembrolizumab (200 mg q3w for up to 35 doses) with continued ART in 30 HIV-positive patients affected by metastatic or locally advanced cancer, including one NSCLC patient. The study showed that this anti PD-1 agent is safe in PLWH in terms of rate and grade of adverse event, comprising immune-related events of clinical interest (irECI), similar for frequency to those occurring in HIV-negative patients treated with the same drug inside pivotal trials. Pembrolizumab did not significantly interfere with ART efficacy, nor in terms of T helper count nor HIV-virus replication control [80].

Waiting for results from other currently ongoing prospective trials of ICIs in this special patient population, some suggestions about this topic come from experience with lower level of evidence.

According to a small retrospective study (*n* = 17), anti PD-1 and anti PD-L1 appeared safe with no detrimental effect on HIV-infection control to treat PLWH with advanced solid tumor, including also NSCLC histology (*n* = 10) [81].

Confirmation in this sense came from another real-life experience from a series of PLWH cancer patients, of whom 21 were affected by NSCLC. The efficacy and safety of anti-PD1 agents nivolumab and pembrolizumab was demonstrated, and no negative effect of immunotherapy on T helper count and/or HIV-viral load was registered [82].

Others small series and case reports highlighted and confirmed that ICIs are safe and active in NSCLC from HIV-positive patients [83,84,85].

Ad-hoc designed clinical trials testing ICIs in cancer patients with HIV-infection are needed to validate the above findings.

## 4. Perspectives

Regardless of immune status, PLWH are in any case much more predisposed to develop cancers because of relevant presence of carcinogenic stimuli, even in terms of frequent co-viral infections, such as high-risk human papillomavirus (hrHPV) and HBV/HCV.

Preventing smoking may potentially avoid near a quarter of NADC in PLWH, according to a wide collaboration of cohort studies [86].

Morbidity and mortality for cancer could be lowered in HIV-positive people, like in the rest of the population, by the adoption of a screening procedure [87]. This might be true also for LC in PLWH, who generally smoke more and tend to quit smoking less than the HIV-negative adulthood counterpart [88].

Based on results from a national screening trial, the U.S. Preventive Services Task Force suggested to screen yearly heavy smokers older than 55 years with a low-dose computed tomography (LD-CT) [88,89].

Recognized thresholds in this specific case were 30-pack years, but maybe could be lowered in people with a history of immunodeficiency. It is unknown whether susceptibility to smoke induced DNA-damage is higher in HIV-positive people rather than in those HIV-negative, but certainly smoke and HIV exert a cooperative inflammatory and immunosuppressive action [17]. A French trial, conducted in PLWH with a nadir T-helper less than 350 cells/mL, showed that a single round LD-CT is able to early detect LC in patients older than 40 years and with at least 20-pack years [90].

Waiting for results from other studies, it would be advisable to offer LC screening to high-risk patients, rather than to all PLWH globally considered. The possible LD-CT false positive findings in HIV-positive patients could in fact be substantial, due to their higher risk of pulmonary infections by different germs (bacteria, mycobacterium, fungi). To combine radiomics with clinical-radiological features may be helpful in identifying those opportunistic pulmonary infections (OPIs) that mimic LC in HIV-infected patients [91].

HIV infection and/or AIDS condition are common exclusion criteria of most cancer clinical studies [92]. The scientific community has to review this aspect, widening the access to trials to those long-marginalized patient categories [93]. Evidence-based medicine is necessary to refine the therapeutic algorithm for PLWH suffering from malignancies; to date, the therapeutic approach has derived from that usually used for HIV-negative patients, but it is conceptually a mistake, because some characteristics may potentially differ based on HIV-status.

No ad-hoc approved and shared guidelines exist to treat PLWH having cancer, and this is quite destabilizing for healthcare providers, therefore find themselves dealing with bias in their approach to the HIV-positive cancer patient [94].

In the recent past, PLWH with cancer have been undertreated with a subsequent negative impact on their prognosis, but no excess in toxicity of antineoplastic agents in HIV-infected patients has shown to date. The best oncologic treatment available should be offered to these patients, taking into account possible DDIs and CD4+ counts. A better global health policy awareness, as well as clinicians’ sensitization and cooperation are fundamental to guarantee a wider access to cancer therapies.

Anyhow, for some cancers, HIV-infected patients live less than those HIV-uninfected, even after adjusting for administered cancer treatments [50]. Despite PLWH with NSCLC, when treated with the best HAART, could have a comparable cancer mortality compared to the general population, they have a significantly higher all-cause mortality: further investigations are needed to understand the reasons of this phenomena, and how HIV worsen mortality in people with cancer and other comorbidities [62].

Clinical trials analyzing the impact of HIV infection on cancer and cancer treatment are urgently warranted, and results of on-going trials will possibly contribute to expanding knowledge on this largely unexplored health issue (Table 2).

## Figures and Tables

**Figure 1 ijms-21-01601-f001:**
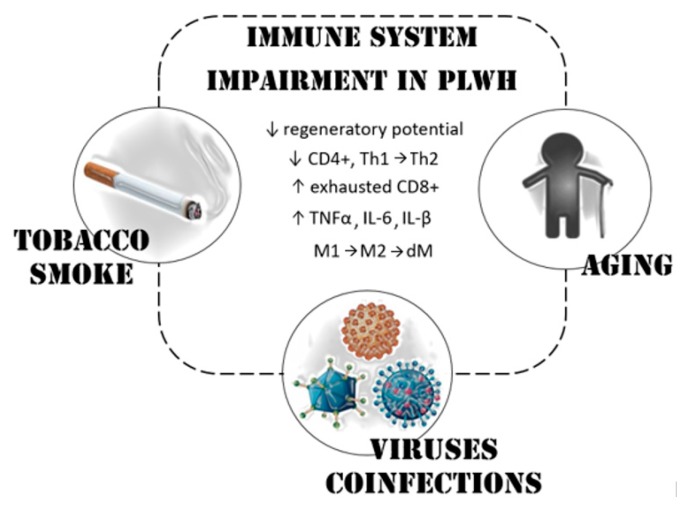
Pathogenesis of LC in people living with HIV (PLWH). Immune system evasion represents the main risk factor for cancer in HIV-positive patients. PLWH constitutively have a strong Th2-humoral oriented immunity, an augmented expression of alternatively activated macrophages (M2) and cytokines responsible of chronic inflammation. These features, together with the impairment of both T lymphocytes pool and function, are linked not only to the HIV infection itself; also, the quite common tobacco smoke habits, oncogenic viruses coinfections (Epstein-Barr virus, hepatitis B virus, hepatitis C virus, Kaposi sarcoma herpesvirus) and immune-senescence of nowadays long-survivor PLWH contribute to their higher risk of LC onset. dM: deactivation of macrophages; M1: classically activated macrophages; Th1: type 1 T helper; Th2: type 2 T helper.

**Figure 2 ijms-21-01601-f002:**
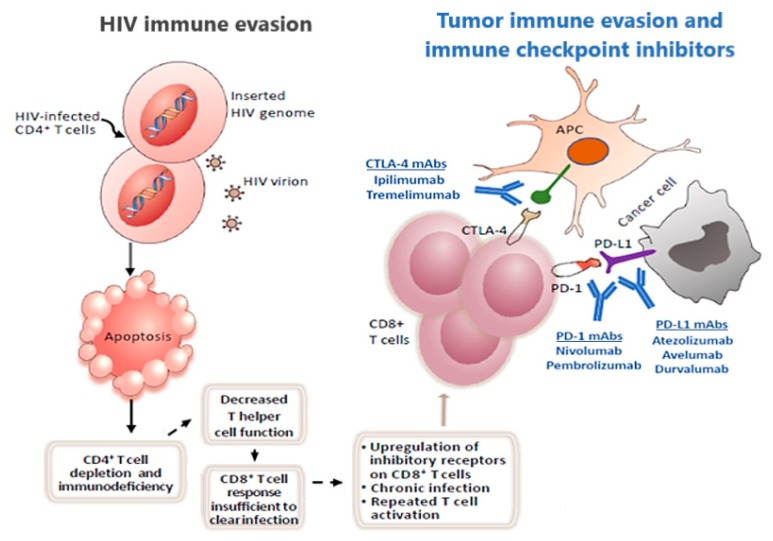
The T cell response to HIV infection, tumor immune evasion in HIV positive patients and immune checkpoint blockade. HIV infection lead both to a depletion and decreased function of CD4+ T cells. This phenomenon causes an insufficient anticancer cytotoxic T cell response and CD8+ T cells to upregulate their inhibitory receptors (iRs), thus increasing cancer risk and tumor immune evasion. Immune checkpoint inhibitors aim to revert cancer immune escape, acting on iRs expressed by tumor cell or by CD8+ T cells, thus that the use of these agents may be crucial for HIV infected people with cancer. CTLA-4: Cytotoxic T-Lymphocyte Antigen 4; mAbs: monoclonal antibody; PD-1: programmed cell death protein 1; PD-L1: Programmed death-ligand 1.

**Table 1 ijms-21-01601-t001:** Epidemiology, clinical–pathological characteristics and prognosis of lung cancer in HIV positive people compared to the general population. The table represents in summary those current available data about lung cancer epidemiology, features and prognosis that may differ between HIV-infected and uninfected individuals. HIV seems to have a prognostic value, meanwhile no clear difference by HIV-status emerges in terms of efficacy of oncologic treatments (see the text).

Lung Cancer Features	Subgroups	HIV + Patients	General Population	References
**Standardized incidence ratio** (**SIR**)	-	2.6 (2.1–3.1)	-	Shiels MS et al. J Acquir Immune Defic Syndr 2009
**Median age at cancer diagnosis**	-	44–52 years	70 years	Shiels MS, et al. Ann Intern Med 2010
**Gender**	male	86%	57.8%	Spano JP et al. Med Oncol 2004
**Race**	white	no current available data		
	black			
**Stage at diagnosis**	stage III or IV	77–100%	75%	Kiderlen TR et al. Oncol Res Treat 2017
**Histological type**	adenocarcinoma	49%	36–50%	Kiderlen TR et al. Oncol Res Treat 2017
	squamous cell carcinoma	20%	27–30%	
	large cell carcinoma	3%	4–18%	
	small cell carcinoma	15%	3–9%	
**Survival rate**	2-year	10%	31%	Biggar RJ et al. J Acquir Immune Defic Syndr 2005
	5-year	10%	19%	Marcus JL et al. Cancer Epidemiol Biomarkers Prev 2015

**Table 2 ijms-21-01601-t002:** Clinical trials investigating HIV and lung cancer. The table listed clinical studies investigating the impact of HIV infection on risk factors, epidemiology, features of LC, as well as implication for LC treatments. Information of active/closed clinical trials has been derived from https://clinicaltrials.gov. The three trials marked in light blue enroll/enrolled people with cancer HIV-uninfected or PLWH without cancer, but anyhow the contribute from these trials might be important to acquire more information about DDIs. **when available. ART: antiretroviral therapy; ChT: chemotherapy; DCR: disease control rate; HL: Hodgkin Lymphoma; ICIs: Immunological Checkpoint Inhibitors; LC: lung cancer; MTD: maximum tolerated dose; NGS: next-generation sequencing; NSCLC: non-small cell lung cancer; ORR: objective response rate; pts: patients; PS: performance status; pts: patients.

Trial NCT (Acronimus)	Drug Tested	Trial Title	Endpoint and Results**	Status
**01296113** (**CHIVA**)	Carboplatin plus pemetrexed	Chemotherapy for LC in HIV+ pts with advanced non-squamous NSCLC	DCR after 4 cycles of carboplatin plus pemetrexed	Completed No results posted
**00276588**	Gemcitabine plus carboplatin followed by paclitaxel	Gemcitabine and carboplatin followed by paclitaxel in pts with PS = 2,3 or other significant co-morbidity (HIV or s/p organ transplantation) in advanced NSCLC	Sequential ChT is well tolerated and active. The survival is comparable to that of other regimens utilized in PS = 2 pts with superior tolerability. The prognosis for these pts is very poor even with treatment	Completed Results posted
**02134886**	Erlotinib	Erlotinib Hydrochloride in treating NSCLC that is metastatic or cannot be removed by surgery in pts with HIV	Safety, tolerability and MTD of erlotinib in combination with ART	Terminated (Poor enrolment)
**01822522**	Cabozantinib	Cabozantinib S-Malate in treating pts with advanced solid tumors and HIV	Safety, tolerability and MTD of cabozantinib	Recruiting
**03304093** (**CHIVA2**)	Nivolumab	Immunotherapy by nivolumab for HIV+ pts with advanced NSCLC	DCR	Recruiting
**03767465** (**PembroHIV**)	ICI	Treatment with ICIs of HIV-infected subjects with cancer (advanced melanoma or other cancers in which the use of ICIs is clinically indicated)	Changes in HIV-viral load and immune-phenotype of cellular populations	Completed No results posted
**02408861**	Ipilimumab plus nivolumab	Nivolumab and ipilimumab in treating pts with HIV associated relapsed or refractory classical HL or solid tumors (comprising LC) that are metastatic or cannot be removed by surgery	MTD of nivolumab, ORR, immune function, change in immune status/HIV viral load	Recruiting
**00791336**	Nelfinavir with RT and ChT	Study to evaluate using nelfinavir with chemoradiation for NSCLC	PCR	Terminated (Poor enrolment)
**01249443**	Paclitaxel plus carboplatin	Paclitaxel and carboplatin in treating pts with metastatic or recurrent solid tumors (comprising NSCLC) and HIV	- Safety, tolerability of vorinostat in combination with ChT. - MTD of the combination	Terminated (Inadequate accrual rate) No results posted
**01567722**	-	Collecting and studying tissue samples from pts with HIV-Associated Malignancies (diffuse large B-cell lymphomas, LC, anal cancer and cervical cancer)	- To obtain high-quality tissue from pts with HIV-1 malignancy - To study clinical, genetic, and immunologic parameters with prognostic significance and/or involved in the initiation/progression of HIV-1 malignancies, including complete NGS of HIV-associated cancers	Recruiting
**01748136** (**NA_00036809**)	-	Screening for LC in the HIV pts	Differences in stage distribution of HIV-seropositive pts at LC diagnosis between those who are screened by spiral CT and historic controls	Completed No results posted
**01207986** (**EP48 HIV CHEST**)	-	Early LC diagnosis in HIV infected population with an important smoking history with low dose CT: a pilot study	Prevalence of LC detected by low-dose CT scan	Completed No results posted
**00491335**	-	HIV infection and tobacco use among injection drug users in Baltimore, Maryland: a pilot study of biomarkers	- To characterize smoking habits and compare tobacco use among HIV-infected and uninfected drug users - To compare serum nicotine levels and spirometry results, as a marker of tobacco use and a marker of damage to lung function, respectively	Completed No results posted
**01447589** (**NelfLung**)	Nelfinavir plus radical radiotherapy	Radical lung radiotherapy plus nelfinavir	MTD of nelfinavir	Withdrawn (poor enrolment )
**00589056**	Nelfinavir with RT and ChT	Nelfinavir, RT, cisplatin and etoposide in treating (HIV-uninfected) pts with stage III NSCLC that cannot be removed by surgery	Nelfinavir administered with concurrent ChT-RT is associated with acceptable toxic effects and a promising ORR, local failure, PFS and OS in unresectable NSCLC	Completed
**03367754**	Pembrolizumab	A single dose of pembrolizumab in HIV-infected people	Safety of pembrolizumab in PLWH who have a low CD4+ T cell count despite taking medicines that control HIV replication	Recruiting
**02595866**	Pembrolizumab	Pembrolizumab in pts with HIV and relapsed/refractory or disseminated malignant neoplasm (comprising NSCLC)	- Frequency of observed AEs - Incidence of immune-related AEs of clinical interest - Incidence of cART-related AEs	Recruiting
**03858491** (**POP-NSCLC**)	Osimertinib	Pharmacokinetic Boosting of Osimertinib in pts with EGFR-mutated NSCLC	Evaluate if systemic exposure of osimertinib is increased when it is co-administered with anti-HIV drug cobicistat	Not yet recruiting
**03706625** (**IDeATIon**)	-	Integrated discovery of new immuno-molecular actionable biomarkers for tumors with immune-suppressed environment	Analyse tumor biomarkers on frozen biopsy for three types of cancer (non-HL, LC and glioma)	Not yet recruiting

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
