# Peer review of "Lung Cancer (LC) in HIV Positive Patients: Pathogenic Features and Implications for Treatment"

_ijms, 2020, doi:10.3390/ijms21051601_

Round 1

Reviewer 1 Report

I read with interest your review. I have just few suggestions:

1) modified the FIg.2 is not so clear

2) Table 1  should be improved

Author Response

We analyzed critically the figure 2, previously the message of the figure itself could be misleading for the readers. Thus, we modified the layout of the text, modified the two main titles, modified the title of the figure caption and added a phrase in the figure caption to clarify the strong interplay between HIV immune suppression, the cancer risk and cancer immune-evasion in HIV patients and the application of immune check point inhibitors in this context. Now, the previously named table 1 is named as table 2, because we just added a new table, to respond to other reviewer comment. After your advise, we added new recent available trials about the topic in the table 2, and also added a column with the drug tested (when indicated).

Reviewer 2 Report

This is about lung cancer in HIV+ patients. And I understand there are not many data on these patients.

Major Commnets

Please describe the differences concisey between HIV+ and - patients with lung cancer. For example, Demographic differnces in table format; the infulence of curative treatment of lung cancer on HIV, and vice versa, etc.

Minor commnets

Line 63, typo There are repeated '1's before many subtitles.

Author Response

We added a new table (table 1) with respective caption, to describe the main differences in LC features between HIV infected and HIV uninfected individuals. We modified line 63 sentence that previously did not sound good in english.

This manuscript is a resubmission of an earlier submission. The following is a list of the peer review reports and author responses from that submission.